# Electrochemical surface passivation of LiCoO$_2$ particles at ultrahigh voltage and its applications in lithium-based batteries

Jiawei Qian[1,2], Lei Liu[3], Jixiang Yang[1,2], Siyuan Li[1,2], Xiao Wang[1,2], Houlong L. Zhuang[3] & Yingying Lu[1,2]

Lithium cobalt oxide, as a popular cathode in portable devices, delivers only half of its theoretical capacity in commercial lithium-ion batteries. When increasing the cut-off voltage to release more capacity, solubilization of cobalt in the electrolyte and structural disorders of lithium cobalt oxide particles are severe, leading to rapid capacity fading and limited cycle life. Here, we show a class of ternary lithium, aluminum, fluorine-modified lithium cobalt oxide with a stable and conductive layer using a facile and scalable hydrothermal-assisted, hybrid surface treatment. Such surface treatment hinders direct contact between liquid electrolytes and lithium cobalt oxide particles, which reduces the loss of active cobalt. It also forms a thin doping layer that consists of a lithium-aluminum-cobalt-oxide-fluorine solid solution, which suppresses the phase transition of lithium cobalt oxide when operated at voltages >4.55 V.

[1] College of Chemical and Biological Engineering, Zhejiang University, Hangzhou 310027, China. [2] State Key Laboratory of Chemical Engineering, Institute of Pharmaceutical Engineering, Zhejiang University, Hangzhou 310027, China. [3] School for Engineering of Matter, Transport and Energy, Arizona State University, Tempe, AZ 85287, USA. Correspondence and requests for materials should be addressed to H.L.Z. (email: hzhuang7@asu.edu) or to Y.L. (email: yingyinglu@zju.edu.cn)

Rechargeable lithium-ion batteries (LIBs) have been used widely in various portable electronics since their first commercialization by Sony Corporation in 1991 and, more recently, in large-scale electrical vehicles (EVs) and energy storage grids (EEGs). Because they are growing rapidly in industrial applications, LIBs are needed that have a higher energy density and greater power output[1–4]. The most prominent cathode materials are based on the crystal structure of layered, spinel, and olivine structures that consist of lithiated cobalt, nickel, and manganese-based oxides, or polyanion materials[3,5].

Among the various cathode materials, lithium cobalt oxide ($LiCoO_2$, LCO) is used presently in >31% of LIBs that are manufactured because of its well-ordered, α-$NaFeO_2$ layered structure, which enables facile scalable production and fast and reversible lithium intercalation[3,6]. Specifically, the LCO has a high theoretical capacity of 274 mAh g$^{-1}$, but the practical discharge capacity is only ~ 140 mAh g$^{-1}$ ($Li_{1-x}CoO_2$, $x \approx 0.5$, ~ 4.2 V vs. Li/Li$^+$)[7,8]. The underlying reason for such a large shortage of capacity is the phase transition from a hexagonal to a monoclinic phase that starts at ~ 4.2 V[8,9]. Extraction of lithium to concentrations more than 0.5 mole leads to structural instabilities that limits the cut-off voltage for lithium deintercalation to 4.2 V. When operating at voltages >4.2 V, the cycling efficiency and discharge capacity of LCO cells decay rapidly.

A close correlation between the loss in capacity and loss of cobalt has been observed[10]. Specifically, as lithium ion (Li$^+$) is removed from the bulk LCO, Cobalt(III) ion (Co$^{3+}$) is oxidized to Cobalt(IV) ion (Co$^{4+}$), which is an unstable oxidation state. The large amount of Co$^{4+}$ is likely to destroy the cathode crystallinity and dissolve into the electrolyte solution, which leads to the irreversible loss of the active transition metal and drastic interfacial side reactions[9,12]. When the voltage reaches 4.55 V, the phase transition from O3 to the H1-3 host occurs together with a significant drop in the c-lattice parameter in bulk LCO particles. Lithium-ion diffusivity decreases consequently, which leads to increased concentration gradients and huge internal strains. The irreversible phase transition degrades the mechanical properties of the bulk LCO crystallites greatly and limits the utilization of LCO particles[10,13].

To promote the cyclability of LCO cells at high capacities, for decades researchers have considered many strategies to protect LCO particles. Of the various approaches, coating LCO particles with metallic compounds (metal-oxide, -phosphates, and -fluorides) or lithium compounds ($LiAlO_2$, $Li_2CO_3$, $LiNi_{0.5}Mn_{1.5}O_4$, etc)[8,14,15] are considered the most promising, because they directly hinder the contact between LCO and the electrolyte solution and, thus, delay structural degradation of the interface. Recently, this method was later promoted into a dual treatment that coating the LCO surface and doping partial element into

the bulk LCO simultaneously to address the problem of structural instability and the formation of microcracks. For example, by applying Mg and P elements to the surface of LCO, the bulk of LCO boosted the retention of cycle capacity from 71.8 to 86.5% after 100 cycles at elevated temperatures with a high cut-off voltage of 4.47 V[8]. In spite of these achievements, research on operating LCO cells >4.55 V has been reported rarely. It is intrinsically difficult to avoid the irreversible phase transition from O3 to H1-3 at voltages >4.55 V, which results in most of the modified LCO cells having a reversible capacity <180 mAh g$^{-1}$ at 0.2 C. If the cut-off voltage is improved from 4.5 V to 4.6 V, an excess initial capacity of ~ 25 mAh g$^{-1}$ could be obtained[16].

Here, we report on the enhanced electrochemical performance of LCO cells at a high cut-off voltage of 4.6 V using a ternary Li, Al, F-based hybrid treatment. The Li, Al, F-modified $LiCoO_2$ (LAF-LCO) is fabricated through a facile and scalable hydrothermal reaction. The resulting Al, F-enriched coating layer with MO (M=Li, Al) nanoparticles effectively resists HF attack from liquid electrolytes and improves the interfacial stability and structural integrity at 4.2 > V[8,17]. Meanwhile, the introduction of excess Li forms a fast Li$^+$ diffusion pathway within the coating layer[15]. It also forms a solid solution that contained Li-Al-Co-O-F beneath the surface coating layer by LAF doping[5], which restrains the harmful irreversible phase transition and enabled reversible cycling at >4.55 V. We calculates the most stable structure of the solid solution when operating at 4.6 V using DFT, and we tentatively propose $Li_{1/3}Al_{1/3}Co_{2/3}O_{4/3}F_{2/3}$ to be the most probable structure. Given all these advantages, the LAF-LCO cells deliver excellent retention of capacity of 81.8% after 200 cycles with high LCO loading ($12.6 \pm 0.3$ mg cm$^{-2}$), although the retention of capacity of bare LCO cells is only 32.8%. We also perform full-cell measurements with the utilization of a compatible amount of graphite as the counter electrode, and we demonstrate that a high specific energy density of 600 Wh kg$^{-1}$ at a material level is obtained after 70 cycles, which offers a promising practical application for high-energy, lithium-based batteries.

## Results

**Formation mechanism of LAF- modified LCO particles**. To synthesize the LAF-LCO cathode material we mixed an aqueous solution of multiple salts of $Al(NO_3)_3 \cdot 9H_2O$, $LiNO_3$, and $NH_4F$, and $Al(NO_3)_3 \cdot 9H_2O$ and $LiNO_3$ were hydrolyzed to $Al(OH)_3$ and LiOH, respectively (Fig. 1a)[18]. Then, bare LCO powder with a particle diameter of 5 μm was added slowly and $Al(OH)_3$, LiOH, and F$^-$ migrated to the surface of the LCO particles[19].

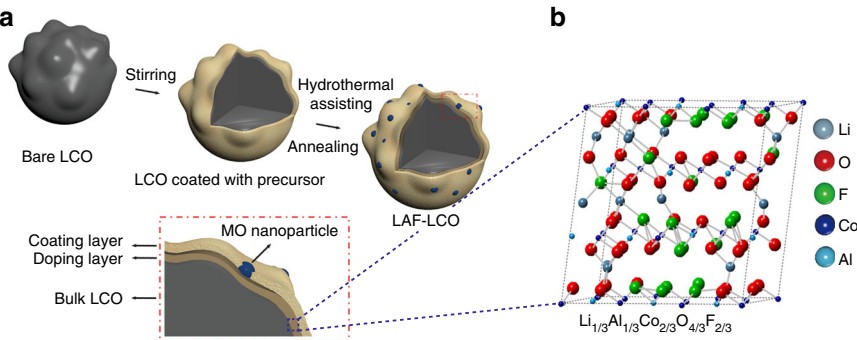

**Fig. 1** The formation and synthetic process for constructing LAF-LCO. **a** Schematic illustration for the synthetic process to produce LAF-LCO. **b** The most probable supercell structure of Li-Al-Co-O-F solid solution using stoichiometric calculations

The chemical reactions were:

$$Al^{3+} + 3H_2O \rightarrow Al(OH)_3 + 3H^+ \quad (1)$$

$$Li^+ + H_2O \rightarrow LiOH + H^+ \quad (2)$$

The resultant solution was then transferred into a Teflon-lined autoclave for hydrothermal reaction, during which partial Al(OH)$_3$ reacted with LiOH and H$_2$O to yield LiAl(OH)$_4 \cdot$H$_2$O[18], and a layer of MOH (M=Li, Al) nanoparticles with negligible fluorides was formed on the surface of the LCO particles. The resultant precursor-coated, LCO particles were annealed at 500 °C, and the final product of LAF-LCO was obtained, which consisted of a fluorine-enriched, protective layer with sparse MO nanoparticles on the surface of LCO and a Li-Al-F doping on the subsurface of LCO. Specifically, during the annealing process, the polycrystalline structure of MO nanoparticles was formed that contained Al$_2$O$_3$ and LiAlO$_2$ that were derived from Al(OH)$_3$ and LiAl(OH)$_4 \cdot$H$_2$O, respectively. The detailed chemical reactions were:

$$Al(OH)_3 + LiOH + H_2O \rightarrow LiAl(OH)_4 \cdot H_2O \quad (3)$$

$$2Al(OH)_3 \rightarrow Al_2O_3 + 3H_2O \quad (4)$$

$$LiAl(OH)_4 \cdot H_2O \rightarrow LiAlO_2 + 3H_2O \quad (5)$$

The formation of the fluorine-enriched, protective layer was demonstrated by TEM (Fig. 2c, d), and the formation of the Li-Al-Co-O-F solid solution in the bulk LCO was predicted by DFT calculation in the next section and characterized by SAED and XPS (Figs 2k, l, Fig. 3). We chose 500 °C as the annealing temperature to obtain both the solid solution phase of Li-Al-Co-O-F and the surface coating layer of MO particles, because Al doping into the bulk LCO began at 500 °C and complete Al doping was achieved at 800 °C[20]. We performed the calcination under the protection of argon gas instead of open air to avoid the formation of superabundant MO nanoparticles that can greatly block Li-ion diffusion.

Understanding the synthesis process enabled us to control the morphology of modified LCO. To study the influence of the concentrations of the precursor solution, we synthesized LAF-LCO with various amounts of LAF, 2% AF-LCO that was formed without LiNO$_3$, and 2% LF-LCO that was formed without Al(NO$_3$)$_3 \cdot$9H$_2$O (Fig. 2a, b, Supplementary Fig. 1). The fluorine-enriched, coating layer with sparsely dispersed MO nanoparticles on the 2% LAF-LCO surface (Fig. 2a–e) was observed using SEM and TEM analyses, and a smooth surface morphology was obtained for bare LCO particles (Supplementary Fig. 1h). For 0.5% LAF-LCO and 1% LAF-LCO, the fluorine-enriched, coating layer was formed with no observation of MO nanoparticles. After the salt concentration increased to 5%, more MO nanoparticles were formed with a larger diameter of ~ 150 nm compared to that on the 2% LAF-LCO surface. The 2% AF-LCO showed fewer MO particles, but it had a larger particle size of ~ 400 nm. In the absence of Al doping (2% LF-LCO), negligible MO nanoparticles were formed, which indicated a strong correlation between the presence of Al$^{3+}$ and the formation of MO. The above results also suggested that as Al$^{3+}$ increased in the salt solution, MO nanoparticles formed and the grain size increased with the increased possibility of self-precipitation of the oxides[21]. Hence, the synthesized nanoparticles of MO were mainly due to the presence of Al(NO$_3$)$_3 \cdot$9H$_2$O.

**Stoichiometric predictions of Li-Al-Co-O-F solid solution in bulk LCO particles.** We performed DFT calculations on the formation of Li-Al-Co-O-F solid solution in the bulk LCO particles and proposed Li$_{1/3}$Al$_{1/3}$Co$_{2/3}$O$_{4/3}$F$_{2/3}$ as the most probable stoichiometry for the solid solution (Fig. 1b). The detailed calculation process is shown in Supplementary Note 1.

We computed the energy change of reaction which forming Li$_{1/3}$Al$_{1/3}$Co$_{2/3}$O$_{4/3}$F$_{2/3}$ and found that this reaction was more strongly exoergic than reaction which forming LiCoO$_2$ (−4.48 vs. −3.69 eV). Because there was no available evidence for the deintercalation of the Li-ion from Li$_{1/3}$Al$_{1/3}$Co$_{2/3}$O$_{4/3}$F$_{2/3}$ at 4.6 V, we proposed a hypothesis that Li$_{1/9}$Al$_{1/3}$Co$_{2/3}$O$_{4/3}$F$_{2/3}$ would be formed with the same amount of Li as Li$_{1/9}$CoO$_2$. We also calculated the formation energy of Li$_{1/9}$Al$_{1/3}$Co$_{2/3}$O$_{4/3}$F$_{2/3}$ in reaction (9) with the structure illustrated in Supplementary Fig. 2d. Similarly, the formation energy became more exoergic and changed from −0.06 eV of Li$_{1/9}$CoO$_2$ to −3.56 eV. Both increases in the formation energies confirmed that forming a Li-Al-Co-O-F solid solution was useful for stabilizing the subsurface structure of LCO.

The $c$-lattice parameters of LiCoO$_2$ (14.35 Å), Li$_{1/9}$CoO$_2$ (13.98 Å), Li$_{1/3}$Al$_{1/3}$Co$_{2/3}$O$_{4/3}$F$_{2/3}$ (14.80 Å), and Li$_{1/9}$Al$_{1/3}$Co$_{2/3}$O$_{4/3}$F$_{2/3}$ (14.60 Å) were obtained using the above methods. The $c$-lattice parameter of LiCoO$_2$ (2.58%) changed more than that of Li$_{1/3}$Al$_{1/3}$Co$_{2/3}$O$_{4/3}$F$_{2/3}$ (1.55%) during deintercalation. Meanwhile, the $c$-lattice parameters of Li$_{1/3}$Al$_{1/3}$Co$_{2/3}$O$_{4/3}$F$_{2/3}$ in the lithiated state and in the delithiated state were larger than that of LiCoO$_2$. The smaller change in $c$-lattice during cycling indicated the stable crystalline structure when using Li$_{1/3}$Al$_{1/3}$Co$_{2/3}$O$_{4/3}$F$_{2/3}$, and the large $c$-lattice parameters of Li$_{1/3}$Al$_{1/3}$Co$_{2/3}$O$_{4/3}$F$_{2/3}$ increased Li diffusivity, undermined ion concentration gradients, and lowered the internal strain[22,23]. Because the capacity fading of LCO was related closely to the internal strain and the consequent structural stability at high voltages >4.55 V, the formation of a stable, Li-ion conductive solid solution by LAF surface doping effectively retarded an irreversible phase transition from O3 to the H1-3 host.

**Morphological and structural analyses of LAF-LCO particles.** TEM images described the detailed structure of 2% LAF-LCO (Fig. 2c, d). The thickness of the coating layer was ~ 7 nm, and grain the size of the MO nanoparticles was 60–100 nm. The ultrathin coating layer that contained Li allowed lithium ions to diffuse freely, which resulted in an enhanced rate capability. High magnification TEM (Fig. 2e) and EDX mapping (Fig. 2f) were used to analyze the structure of the LAF protective layer. MO nanoparticles were distributed sparsely around the surface of the LAF protective layer, and only Al and O elements were detected with no Co and F signals. This result confirmed that no fluoride existed in the MO nanoparticles, which was consistent with the above-mentioned mechanism.

Explicit evidence for the formation of the solid solution induced by LAF coating was provided by comparative high resolution TEM (HRTEM) analysis of the pristine LCO and LAF-LCO. Selected area electron diffraction (SAED) patterns of the primary particles that comprised the pristine sample matched those of other LCO particles that had a layered phase and a rock-salt phase. This may have formed because of the removal of Li from the surface crystal structure, which resulted from the unavoidable formation of poorly conductive Li$_2$CO$_3$ or LiOH when exposed to air or moisture. In the case of LAF-coated LCO particles, the polycrystalline phase was found in region 3, which coincided with the presence of Al$_2$O$_3$ and LiAlO$_2$ in the MO particles. LiAlO$_2$ crystals have three different structures: the hexagonal α-LiAlO$_2$, monoclinic β-LiAlO$_2$, and tetragonal γ-

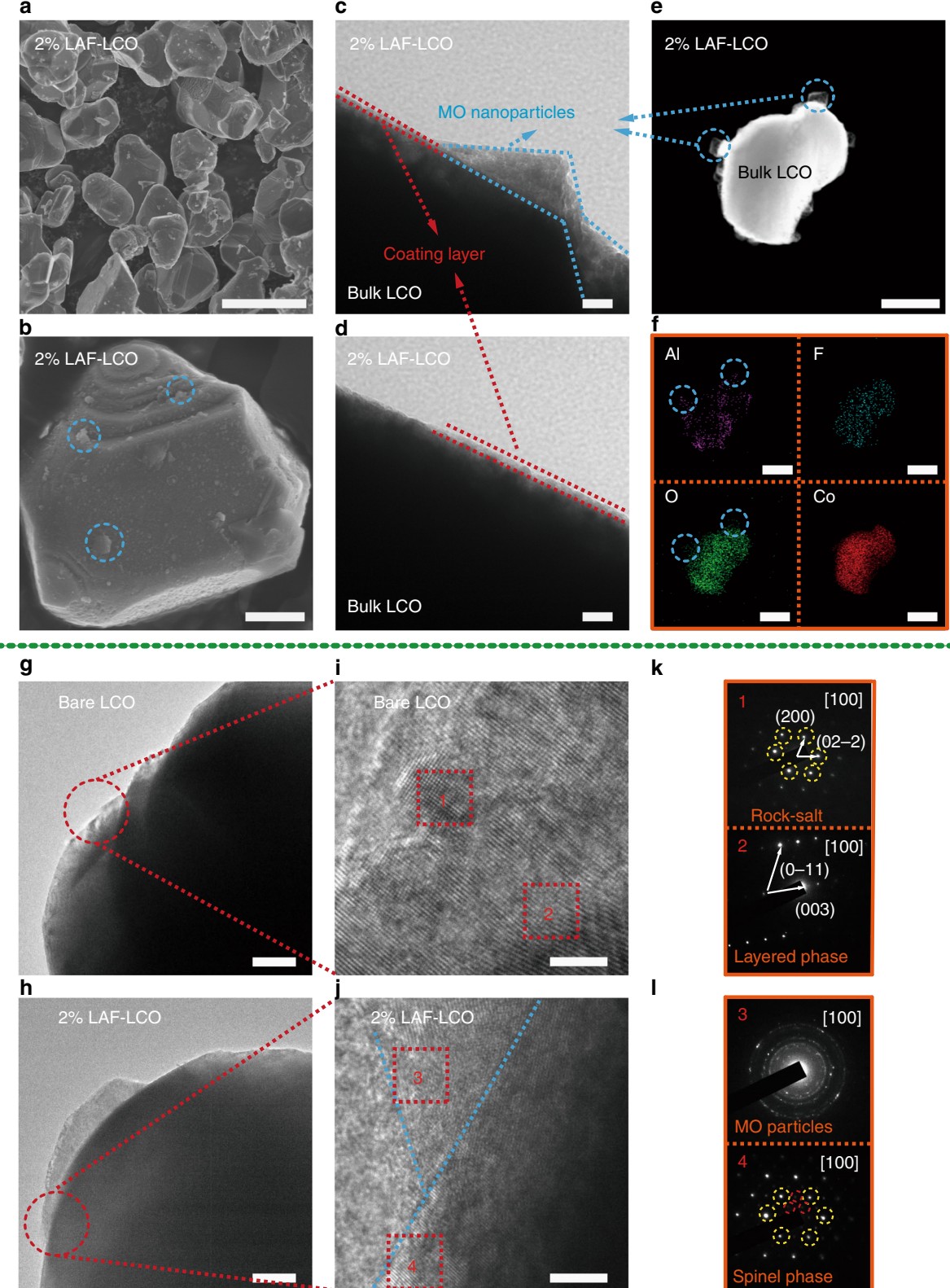

**Fig. 2** Morphological and structural analyses of LAF-LCO. **a**, **b** SEM images of 2% LAF-LCO particles at different magnifications. Scale bar: 5 μm and 2 μm, respectively. **c**, **d** TEM images of 2% LAF coating layers and MO nanoparticles. Scale bar: 20 nm. **e** High magnified TEM image of 2% LAF-LCO with MO nanoparticles. Scale bar: 1 μm. **f** EDS elemental maps of Al, F, O, and Co, which corresponds to **e**. Scale bar: 1 μm. **g**, **h** High magnified TEM images of bare LCO and 2% LAF-LCO, respectively. Scale bar: 200 nm. **i**, **j** High resolution TEM images of the selected regions in **g**, **h**, respectively. Scale bar: 5 nm. **k**, **l** Electron diffraction patterns of marked region 1 and 2 in **i**, and 3 and 4 in **j**, respectively

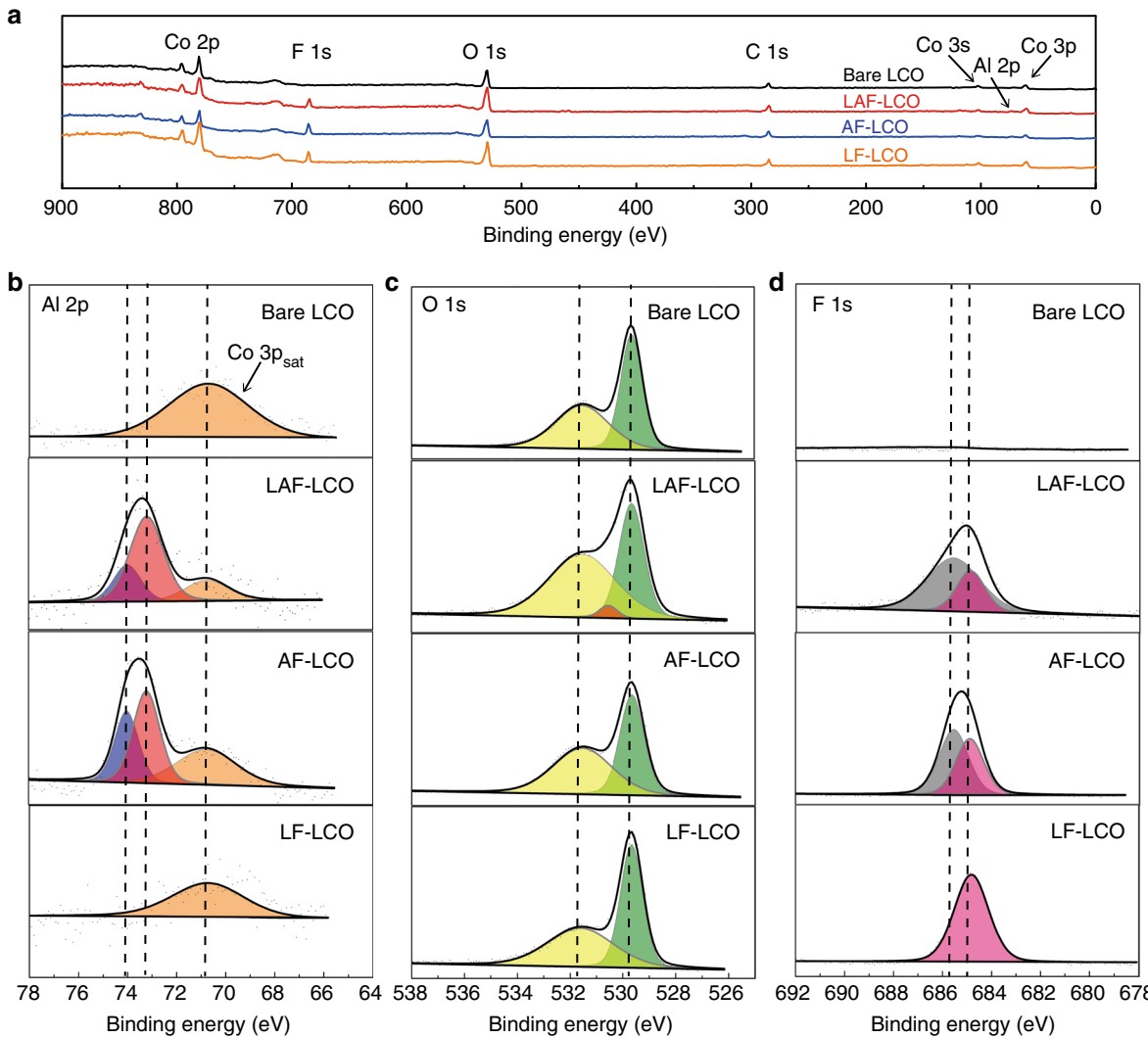

**Fig. 3** Characterization of XPS. **a** XPS patterns of bare LCO and 2% LAF/AF/LF-LCO. **b–d** Fine XPS scans of Al 2p peaks (including Co 3p$_{sat}$ peak), O 1s peaks, and F 1s peaks, respectively. Source data are provided as a Source Data file

LiAlO$_2$[24]. The three crystal structures can be generated in the synthesis process, which leads to the formation of complicated MO structures. Instead, the modified LCO surface presented a uniform spinel phase in region 4[10,25], which resulted from the partial doping of Li, Al, and F in the bulk LCO.

To further establish the possibility of the formation of the solid solution on the surface of bulk LCO, LAF/AF/LF-LCO, and bare LCO samples were analyzed by powder X-ray diffraction (XRD, Supplementary Fig. 4) and X-ray photoelectron spectroscopy (XPS, Fig. 3). By comparing the XRD patterns of modified-LCO with that of bare LCO, the surface coating cannot be detected due to a limited amount of coating. This indicated that the bulk-layered crystalline structure of LCO was not impaired by the hybrid surface treatment. The Co 3p$_{sat}$ peaks (orange peaks, Fig. 3b) were observed at 70.9 eV[20,26], which was near the Al 2p peaks at 73.5 eV. The blue peak, at 74.1 ± 0.1 eV, corresponded to the Al atoms in an oxygen environment, which demonstrated the existence of Al$_2$O$_3$ (74.3 eV) and LiAlO$_2$ (74.1 eV) in the MO nanoparticles. Another component displayed in pink was observed at a lower binding energy of 73.3 eV, which corresponded to the Al atoms in the solid solution[20]. When we compared the area of the blue peak and the pink peak in LAF-LCO and AF-LCO (73.17% of the entire Al 2p peak area in LAF-LCO, 60.49% in AF-LCO), we found that more solid

solution was formed in LAF-LCO. For the O 1s spectra, the green peak at 529.7 eV was characteristic of O atoms in the LiCoO$_2$ (or solid solution) crystal lattice. The yellow peak at higher binding energy of 531.7 eV was associated with oxygen-containing adsorbed species, which can result from the reaction of LCO particles in open air or O atoms on the extreme surface of LiCoO$_2$ with a deficient coordination[20,26]. The O 1s peaks of Al$_2$O$_3$ and LiAlO$_2$, which are usually located at 531.7 eV and 531.0 eV, respectively, merged with the yellow peak[16,20]. The binding energy of LiAlO$_2$ was a little smaller than that of Al$_2$O$_3$, due to the lower electronegativity of Li compared to Al[15,27]. The F 1s peak (purple peak) at 684.8 eV was expected to be the F 1s peak in LiF (684.9 eV)[27], but the gray peak at 685.7 eV observed in LAF-LCO and AF-LCO was likely to be the F 1s peak in the LiAlF$_4$ (685.5 eV)[27]. This indicated that the Li in LCO participated in the reaction to form a Li-Al-F compound on the surface of AF-LCO.

**Electrochemical performance of Li/LCO half cells.** Figure 4a reports the cycling performance of bare LCO and LAF-LCO electrodes as a function of the amount of LAF. Galvanostatic measurements were conducted using Li/LCO cell configuration in the range of 3.0–4.6 V (vs. Li$^+$/Li) at 27.4 mA g$^{-1}$ (at 13.7 mA g$^{-1}$ in the initial two cycles for SEI formation).

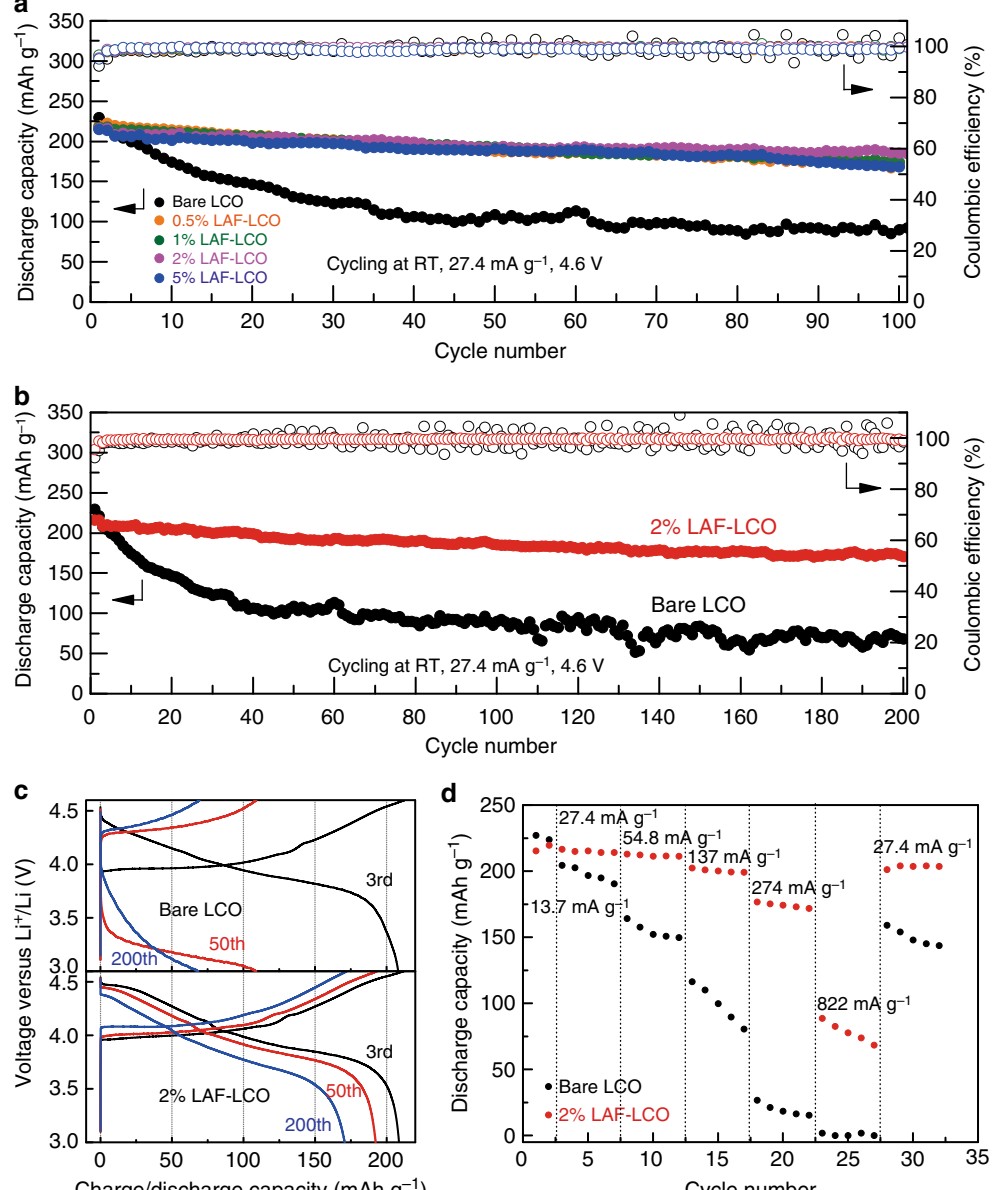

**Fig. 4** Electrochemical performance of bare Li/LCO cells and Li/LAF-LCO cells. **a** Cycling performance of half-cells with bare LCO or LAF-LCO electrodes at room temperature in the voltage range of 3.0–4.6 V (vs. Li$^+$/Li) at current density of 27.4 mA g$^{-1}$. **b** Long-term cycling performance of half-cells with bare LCO or 2% LAF-LCO electrodes at room temperature in the voltage range of 3.0–4.6 V (vs. Li$^+$/Li) at current density of 27.4 mA g$^{-1}$. **c** Discharge-charge profiles of half-cells with bare LCO or 2% LAF-LCO electrodes at different cycles in the voltage range of 3.0–4.6 V (vs. Li$^+$/Li) at current density of 27.4 mA g$^{-1}$. **d** Rate performance of half-cells with bare LCO and 2% LAF-LCO electrodes at room temperature in the voltage range of 3.0–4.6 V (vs. Li$^+$/Li). All cells were pre-cycled for two cycles at a low current density of 13.7 mA g$^{-1}$. Source data are provided as a Source Data file

It was apparent that cells that used LCO with any amount of LAF protection outperformed those with bare LCO. The results for cells with 2% LAF-LCO electrodes were particularly remarkable, because they delivered high capacity retention of 89.1% (185.3 mAh g$^{-1}$) even after 100 cycles.

However, cells with 0.5% LAF-LCO, 1% LAF-LCO, 5% LAF-LCO, and bare LCO delivered 77.6% (170.7 mAh g$^{-1}$), 81.0% (174.2 mAh g$^{-1}$), 81.6% (168.7 mAh g$^{-1}$), and 44.5% (92.6 mAh g$^{-1}$), respectively. The inferior cycling stability of 0.5% LAF-LCO and 1% LAF-LCO compared to 2% LAF-LCO may have been due to an incomplete coating of LAF. For 5% LAF-LCO, the coating may have been so thick with MO particles with poor electro-conductivity that it impeded Li-ion diffusion and eventually led to capacity fading. Cells that used 2% LAF-LCO maintained an

ultrahigh capacity of 170.7 mAh g$^{-1}$ after 200 cycles (18.2% decay over 200 cycles or 0.091% per cycle) (Fig. 4b). For cells with bare LCO, the capacity was only 68.2 mAh g$^{-1}$ after 200 cycles (67.2% decay over 200 cycles or 0.336% per cycle). The Coulombic efficiency (CE) was also monitored during cycling, which is defined as the ratio of discharge capacity to charge capacity. Cells with 2% LAF-LCO showed an average CE of 99.64% after five cycles and cells with bare LCO had an average CE of 98.86%, which indicated that irreversible side reactions occurred during cycling at a high cut-off voltage.

Side reactions between LCO, especially at an unstable high oxidation state Co$^{4+}$, and liquid electrolytes have been one of the key mechanisms for failure of LiCoO$_2$ at a high cut-off voltage. The highly reactive Co$^{4+}$ can destroy the cathode crystallinity,

dissolve in liquid electrolytes, and accelerate electrolyte decomposition. Therefore, it is important to create a protection layer with excellent chemical and electrochemical stabilities for LCO particles to impede side reactions at high voltages[8,12]. The corresponding discharge-charge voltage profiles of cells with bare LCO or 2% LAF-LCO electrodes were generally consistent with the characteristic profiles of Li/LCO in ethyl carbonate/diethylene carbonate (EC/DEC, volume:volume = 1:1) electrolytes (Fig. 4c)[15].

The discharge capacity (208.1 mAh g$^{-1}$) of bare LCO was similar to that (208.6 mAh g$^{-1}$) of 2% LAF-LCO after two cycles for SEI formation, although it delivered a higher initial discharge capacity (229.8 mAh g$^{-1}$) than 2% LAF-LCO (216.2 mAh g$^{-1}$). This can be attributed to the serious destruction of the surface crystal structure during SEI formation without the protective coating materials. Two potential plateaus of the discharge curve were observed at 4.41 and 3.81 V. The short plateau at 4.41 V was attributed to the transition from the H1-3 phase to the O3 phase[9,11], and the long plateau at 3.81 V was attributed to the transition between the two O3 phases[11,28]. After 50 cycles, the short plateau of bare LCO disappeared, which probably resulted from the irreversible phase transition between the O3 and H1-3 hosts in LCO. In contrast, the short plateau in 2% LAF/LCO cells still remained after 200 cycles, which suggested that the irreversible phase transition was restrained.

Figure 4d shows the rate capabilities of cells with bare LCO or 2% LAF-LCO. The 2% LAF-LCO electrode delivered reversible specific capacity of 213.1, 211.9, 200.5, 174.2, and 78.3 mAh g$^{-1}$ at 0.276, 0.552, 1.38, 2.76, and 8.28 mA cm$^{-2}$, respectively (and 27.4, 54.8, 137, 274, and 822 mA g$^{-1}$, respectively). The bare LCO electrode showed reversible specific capacity of 197.8, 154.9, 99.3, 19.7, and 0.7 mAh g$^{-1}$ at 0.276, 0.552, 1.38, 2.76, and 8.28 mA cm$^{-2}$, respectively (and 27.4, 54.8, 137, 274, and 822 mA g$^{-1}$, respectively). With different charge/discharge rates, the LAF/LCO electrode showed higher and more stable specific capacity than that of bare LCO electrodes, which indicated enhanced stability of the LAF-LCO electrode at high voltages. After returning to 27.4 mA g$^{-1}$ after high rate cycling, 95% capacity was recovered for the 2% LAF-LCO electrode.

We also tested the rate capability of 2% LF-LCO and 2% AF-LCO to further confirm the superiority of 2% LAF-LCO (Supplementary Fig. 12). Cells with 2% LF-LCO delivered inferior capacity in the first 22 cycles among the three modified LCO cells because of the poor conductivity of LiF. It is interesting to note that a higher capacity was obtained for 2% LF-LCO cells than for 2% AF-LCO cells during 28–32 cycles. This can be attributed to the destruction of the interfacial protective layers of LF-LCO particles during the initial few cycles that provided additional pathways for Li diffusion. When returning to 27.4 mA g$^{-1}$ after 27 cycles, CE of 95.6% was delivered for 2% AF-LCO cells or 87.5% for 2% LF-LCO cells. The low CE of 2% LF-LCO cells when cycled back to the initial current density indicated the destruction of the interfacial protective layers for 2% LF-LCO particles during cycling. The 2% AF-LCO cells provided enhanced stability, but rendered poor rate capability, which was probably due to its low Li-ion conductivity.

Figure 5a and b report cyclic voltammetry (CV) curves of Li/LCO cells with or without 2% LAF at a scanning rate of 0.1 mV s$^{-1}$. During the initial discharge, similar reduction peaks were observed in both cases at potentials around 3.8 V, which corresponded to the phase transition between two O3 phases, around 4.0 and 4.1 V. This was characteristic of the order–disorder phase transitions[28], and 4.4 V represented the transition from the H1-3 to the O3 phase[29]. The dQ/dV curves in

Supplementary Fig. 13c, d showed similar results. The redox peaks of bare LCO were broader and lower than those of 2% LAF-LCO, which indicated that 2% LAF-LCO possessed a faster kinetic for Li-ion diffusion across the surface of the cathode material. The retarded Li-ion diffusion on the surface of bare LCO could be due to the unavoidable formation of poorly conductive Li$_2$CO$_3$ or LiOH when exposed to air or moisture. Compared with the cathode without LAF protection, which already showed noticeable changes in peak positions and current amplitudes by the 10th cycle, cells with LAF-LCO electrodes exhibited many fewer changes in redox peaks. Remarkably, these redox peaks were still observed even after 200 cycles (Supplementary Fig. 5). The disappearance of the redox peaks at 4.4 and 4.6 V indicated an irreversible transition from the O3 to the H1-3 phase, and the pair of redox peaks at 3.85 and 4.1 V became much smaller, they also shifted much further apart from large polarization due to the formation of harmful resistive interlayers. This means that the introduction of the LACOF doping layer did not change the phase transition behavior of LCO, but it restrained the harmful irreversible phase transition between the H1-3 and O3 host.

The impedance spectra for cells with bare LCO or 2% LAF-LCO electrodes at different cycles are presented in Supplementary Fig. 6. The semicircle in the medium frequency region represented the interfacial resistance between the electrolyte and the electrode[27]. Although the initial semicircles were similar for both bare LCO and 2% LAF-LCO, the semicircle of bare LCO increased significantly when cycling, which indicated sluggish charge transfer at the electrolyte/electrode interface, LAF-LCO remained almost constant and small after extended cycling. To demonstrate that the failure mechanism for the LCO at high cut-off voltages was related to the loss of Co, the Co content in the electrolyte at various cycle numbers was analyzed using an inductively coupled plasma (ICP) technique (Supplementary Fig. 7). As expected, the Co content in cells with the 2% LAF-LCO electrode remained low, although the Co content continuously increased in cells with the bare LCO electrode. This suggested there was less irreversible loss of active Co in Li/LAF-LCO cells.

Post-mortem SEM analyses were performed at different cycles to monitor the surface morphologies of LCO and 2% LAF-LCO electrodes. Figure 5c, d are SEM images of bare LCO and 2% LAF-LCO electrodes before cycling, at 5th cycle and 100th cycle with different magnifications, respectively. It can be seen that after five cycles many LAF-LCO particles were covered with a protective layer, but most of the bare LCO particles showed similar morphologies to those before cycling, which indicated limited SEI formation. After 100 cycles, the protective layer on LAF-LCO particles still remained integrated and well-distributed (Fig. 5c, d, Supplementary Fig. 15). In contrast, the LCO particles without LAF contained visible micro-cracks and an exposed layer structure (circled in red), which suggested there was degradation of the surface structure of the LCO particles. The direct exposure of bare LCO particles to electrolytes and a lack of favorable SEI protection greatly reduced the utilization of active cathode materials after extended cycling at high voltages. To further analyze the evolution of surface structure, the evolution of (003) and (015) peaks of bare LCO or 2% LAF-LCO before cycling and after 100 cycles at 137 mA g$^{-1}$ has been examined in Supplementary Fig.14. The (003) and (015) peaks of bare LCO have shifted apparently to the higher 2$\theta$ angle. This was attributed to the collapse of the crystal structure due to irreversible phase transition[30]. Meanwhile, the (003) and (015) peaks of 2% LAF-LCO barely shifted, which meant that the irreversible phase transition was retarded, and the crystal structure was stable.

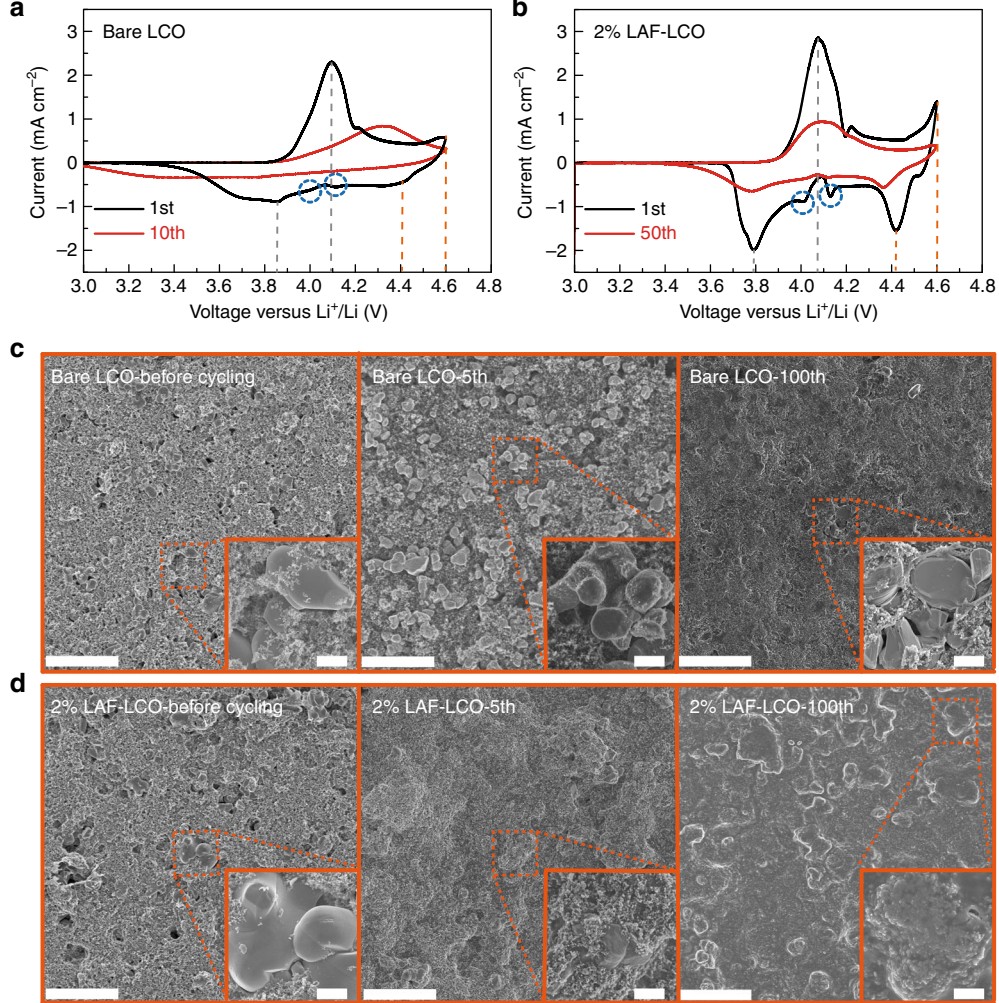

**Fig. 5** Electrochemical and morphology analyses of bare LCO and 2% LAF-LCO electrodes. **a**, **b** Cyclic voltammograms of cells with bare LCO or 2% LAF-LCO electrodes at a scan rate of 0.1 mV s$^{-1}$ in the voltage range of 3.0–4.6 V (vs. Li$^+$/Li). **c**, **d** Top-view SEM images of bare LCO and 2% LAF-LCO electrodes before cycling and after 5 and 100 cycles at room temperature in the voltage range of 3.0–4.6 V (vs. Li$^+$/Li) at 27.4 mA g$^{-1}$. Scale bar: 20 μm and 2 μm. Source data are provided as a Source Data file

The electrochemical performance of 2% LAF-LCO at a higher current density of 137 mA g$^{-1}$ (1.38 mA cm$^{-2}$), or higher voltages (4.65 V or 4.7 V), was also tested (Supplementary Fig. 8, Supplementary Fig. 9, respectively). Cells using 2% LAF-LCO electrodes delivered an ultrahigh capacity of 158.8 mAh g$^{-1}$ after 100 cycles (20.6% decay over 100 cycles or 0.206% per cycle) at 137 mA g$^{-1}$. For bare LCO, the capacity was only 30.3 mAh g$^{-1}$ after 100 cycles (82.2% decay over 100 cycles or 0.822% per cycle). To push the capacity limit, galvanostatic measurements were conducted in the voltage range of 3–4.65 V or 3–4.7 V. Cells with LAF-LCO showed stable cyclability with enhanced capacity, especially during extended cycling.

In light of these results from quantitative analyses, the enhanced electrochemical performance of cells with 2% LAF-LCO electrodes was mainly attributed to the retardation of side reactions and the suppression of Co dissolution by the stable, fluorine-enriched, coating layer. The irreversible transition from the O3 to the H1-3 phase at voltages can also be partially undermined by LAF superficial doping. We summarized different research strategies in the surface coating of LiCoO$_2$ at a high cut-off voltage of 4.6 V (Supplementary Table 1). In comparison with their results, the electrochemical performance by our hybrid, LAF-based, surface treatment was at the superior of the high-voltage LCO field.

**Electrochemical performance of graphite-LCO full cells**. For potential scalable production, full cells with bare LCO or 2% LAF-LCO cathodes and commercial, synthetic artificial graphite (SAG) anodes in the (negative/positive) N/P ratio of 1:1 were assembled and cycled at room temperature in the voltage range of 3.0–4.6 V (vs. SAG) at 27.4 mA g$^{-1}$ (0.276 mA cm$^{-2}$). To the best of our knowledge, this is the first research that demonstrates the high-voltage (4.6 V) electrochemical performance of modified LCO electrodes in full-cell configuration.

As shown in Fig. 6a, the initial CE of graphite/2% LAF-LCO cells was 80.2%, which was much higher than that of graphite/bare LCO cells (67.2%), this was a higher initial capacity of 204.2 mAh g$^{-1}$ compared to 178.3 mAh g$^{-1}$ of bare LCO cells. However, the initial CEs of both full cells were lower than that of half cells, which could be attributed to the irreversible loss of active Li during the SEI formation on electrodes. After cycling, the capacity in graphite/bare LCO cells faded quickly and delivered only 98.4 mAh g$^{-1}$ with an average CE of 98.77% after 70 cycles. In contrast, graphite/2% LAF-LCO cells showed enhanced electrochemical performance with a capacity of 155.6 mAh g$^{-1}$ and an average CE of 99.12%. The observed difference in capacity retention between Li/2% LAF-LCO half cells (92.0%) and graphite/2% LAF-LCO full cells (78.4%) after 70 cycles were

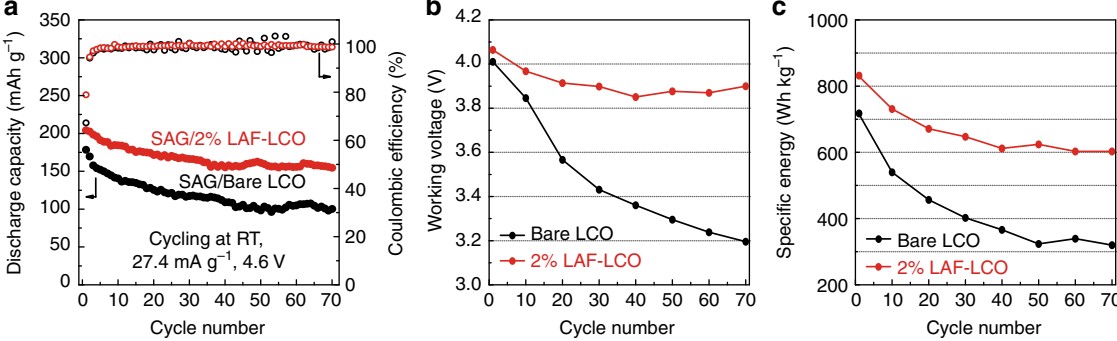

**Fig. 6** Electrochemical performance of synthetic artificial graphite (SAG)/bare LCO or 2% LAF-LCO full cells. **a** Cycle performance of full-cells with bare LCO or 2% LAF-LCO electrodes at room temperature in the voltage range of 3.0–4.6 V (vs. graphite) at a current density of 27.4 mA g$^{-1}$. **b**, **c** Working voltage and specific energy of full-cells with bare LCO or 2% LAF-LCO electrodes. Source data are provided as a Source Data file

associated with the active Li loss that was induced by the decomposition of carbonate liquid electrolytes at 4.6 V[31] and the limited amount of Li in electrodes. The working voltage of cells with 2% LAF-LCO electrodes remained stable around 3.9 V after 20 cycles, but the working voltage with bare LCO electrodes dropped continuously to 3.2 V over 70 cycles (Fig. 6b). It was also interesting that the working voltage of Li/2% LAF-LCO half cells delivered a similar value of 3.9 V, which indicated that the structure of 2% LAF-LCO remained stable when applied in full-cell configuration (Supplementary Fig. 10). We calculated and summarized specific energy, which is defined as specific energy = discharge capacity × working voltage (Fig. 6c). As expected, a high and reversible energy density of 600 Wh kg$^{-1}$ at material level was obtained in 2% LAF-LCO full-cells, but bare LCO full-cells delivered only half of that of graphite/2% LAF-LCO.

For the next level of research standards for practical applications, pouch-type full cells with 2% LAF-LCO electrodes were assembled. The open-circuit voltage of pouch-type full cells with 2% LAF-LCO electrodes after SEI formation and first charge was about 4.51 V, which was similar to that of coin-type half cells (4.54 V) (Supplementary Fig. 16). In addition, we successfully applied the graphite/2% LAF-LCO pouch-cells in a hand-held electric fan, which usually uses three triple-A cells with a working voltage of 4.5 V. The hand-held electric fan worked well using the pouch-type full cell with 2% LAF-LCO electrodes which demonstrated its potential scale-up application (Supplementary Fig. 16).

## Discussion

LiCoO$_2$, as a popular commercial cathode in LIBs, delivers only half of its theoretical capacity at current commercial use. When increasing the cut-off voltage to release more Li out of LCO and increase the utilization of active materials, it is understood that solubilization of Co in the electrolyte and structure disorder of LCO particles occur severely, especially over 4.55 V, leading to rapid capacity fading and limited cycle life. We note that the structural degradation of LCO particles, which resulted from irreversible phase transition from O3 to H1-3 host, is the main reason that limits its application over 4.55 V. Therefore, forming a stable and Li-ion conductive surface layer can effectively retard such irreversible phase transition from O3 to H1-3 host.

We successfully created a stable and conductive LAF-based, composite protective interlayer on bulk LCO particles using a facile and scalable hydrothermal-assisted, hybrid surface treatment. Our DFT calculations predicted the most likely structure of the solid solution on the LCO surface, Li$_{1/9}$Al$_{1/3}$Co$_{2/3}$O$_{4/3}$F$_{2/3}$, and this revealed its excellent thermodynamic stability. The major

challenges for LCO at high cut-off voltages were overcome by our approachwhere undesirable interfacial side reactions were retarded by employing a stable interlayer, and irreversible phase transition over 4.55 V was alleviated by the LAF-doped solid solution. Ultrahigh-energy-density, lithium-based batteries with excellent electrochemical performance were achieved with a high cut-off voltage of 4.6 V. Li/2% LAF-LCO cells showed excellent long-term cycling performance with a capacity of 170.7 mAh g$^{-1}$ at 27.4 mA g$^{-1}$ (0.276 mA cm$^{-2}$) current density and a capacity retention of 81.8% after 200 cycles. When applied in full-cell configuration, graphite/2% LAF-LCO cells delivered a high specific energy density of 600 Wh kg$^{-1}$ at material level after 70 cycles. Our approach is promising for realizing scalable industrial production of high-energy-density, lithium-based batteries.

## Methods

**DFT calculations of Li-Al-Co-O-F solid solution in the bulk LCO.** All the density functional theory calculations were performed using the Vienna Ab initio Simulation Package (VASP)[32]. We used the Perdew–Burke–Ernzerhof functional[33] to describe exchange-correlation interactions between electrons. We also used the potential datasets generated according to the projector augmented-wave method[34,35], where the $1s^2 2s$ states of Li, $2s^2 2p$ states of Al, $3s^2 3p^6 3d^7 4s^2$ states of Co, $2s^2 2p^4$ states of O, and $2s^2 2p^5$ states of F were treated as valence electrons. Plane wave basis sets were used with a cut-off energy of 600 eV. An effective $U$ parameter of 3.32 eV[36] was adopted to deal with the repulsive Coulomb interactions between Co $d$ orbitals following Dudarev's approach[37]. We used the cluster expansion and Special Quasirandom Structures (SQS) methods as implemented in the ATAT package[38]. The $k$-point sampling grids were set to $16 \times 16 \times 16$, $6 \times 6 \times 18$, $20 \times 20 \times 4$, $8 \times 6 \times 2$, $16 \times 16 \times 16$, $2 \times 6 \times 2$, $1 \times 1 \times 1$, and $2 \times 6 \times 2$ for 2-atom Li, 24-atom CoO$_2$, 12-atom LiCoO$_2$, 28-atom Li$_{1/9}$CoO$_2$, 4-atom Al, 2-atom F$_2$, 90-atom Li$_{1/3}$Al$_{1/3}$Co$_{2/3}$O$_{4/3}$F$_{2/3}$, and 84-atom Li$_{1/9}$Al$_{1/3}$Co$_{2/3}$O$_{4/3}$F$_{2/3}$ cells, respectively. All atomic coordinates along will the lattice constants of the crystals were optimized fully until the force convergence criterion of 0.01 eV Å$^{-1}$ was reached.

**Syntheses of LAF/AF/LF-LiCoO$_2$.** To synthesize the LAF/AF/LF-LiCoO$_2$, Al (NO$_3$)$_3$·9H$_2$O (99.99%, Aladdin), LiNO$_3$ (99.99%, Aladdin), NH$_4$F (99.99%, Aladdin), and LiCoO$_2$ (99.8%, Aladdin) were used without further purification as received. LAF-LiCoO$_2$, labeled 0.5% LAF-LCO (0.5 wt% LAF), 1% LAF-LCO (1 wt % LAF), 2% LAF-LCO (2 wt% LAF), 5% LAF-LCO (5 wt% LAF), AF-LiCoO$_2$, labeled 2% AF-LCO (2 wt% AF), LF-LiCoO$_2$, and labeled 2% LF-LCO (2 wt% LF), were synthesized by a facile hydrothermal-assisted method and calcination. The calculated amount of Al(NO$_3$)$_3$·9H$_2$O, LiNO$_3$, and NH$_4$F were dissolved completely in 30 mL deionized water after stirring for 60 min (for example, in the case of forming 2% LAF-LCO, Al(NO$_3$)$_3$·9H$_2$O:LiNO$_3$:NH$_4$F = 0.0683 g:0.0126 g:0.0283 g). Then 1 g of LiCoO$_2$ was added slowly to the solution, and the resultant solution was further stirred for 1 h at room temperature to obtain a miscible mixture of LiCoO$_2$ and precursor solution. The mixture was then transferred into a Teflon-lined autoclave, and the hydrothermal reaction was conducted at 160 °C for 5 h. The product was washed three times with deionized water and then dried in a convection oven at 80 °C for 8 h. The precursor-coated powder was collected and calcined at 500 °C for 6 h in argon gas with a heating rate of 5 °C min$^{-1}$ from room temperature to yield the final product.

**Half-cell fabrications**. The positive electrodes were fabricated with 80 wt% active material, 10 wt% super P as conductive additive, and 10 wt% polyvinylidene fluoride (PVDF, 99.5%, Arkema) as binder. The calculated amount of powder was dissolved in *N*-methyl-1,2-pyrrolidone (NMP, 99.9%, MTI corporation KJ GROUP) to obtain a homogeneous slurry. The slurry was cast onto aluminum foil using the doctor blade technique. The NMP solvent was evaporated at 110 °C for 12 h in a vacuum oven. Coin cells (CR2032) were assembled with LAF/AF/LF-LCO or bare LCO as cathode material, lithium metal as counter electrode, polypropylene (PP, Celgard 2400) as separator, and 1 M $LiPF_6$ in ethyl carbonate/diethylene carbonate (EC/DEC, volume:volume = 1:1, 45 μL) as electrolyte in an argon-filled glovebox. The loadings of active cathode materials in all cells were kept $12.6 \pm 0.3$ mg cm$^{-2}$.

**Full-cell fabrication**. The preparation of cathode materials was the same as that of half-cells. The negative electrode was fabricated with 92 wt% SAG (99.5%, 330 mAh g$^{-1}$, MTI corporation KJ GROUP), 3 wt% acetylene black (99.5%, MTI corporation KJ GROUP) as conductive additive, and 5 wt% PVDF as binder. The obtained slurry was cast onto copper foil using the doctor blade technique, then it underwent the same process as the cathode. The coin-type full cells with a 1.1:1 capacity ratio of negative electrode capacity/positive electrode capacity (N/P), were assembled using the same method as that of half-cells, and the same electrolyte was used. The loading level of negative electrodes was $8.1 \pm 0.3$ mg cm$^{-2}$. The pouch-type full cells with a 1.1:1 N/P capacity ratio with dimensions of 5.6 cm × 4.3 cm were assembled using pouch-cell production machines (MTI corporation KJ GROUP).

**Electrochemical measurements**. The galvanostatic charge–discharge measurements were carried out in the potential range of 3–4.6 V, 3–4.65 V, or 3–4.7 V (vs. Li/Li$^+$) at room temperature with an eight-channel LAND battery tester. The electrochemical impedance spectroscopy (EIS) and CV were tested in an electrochemical workstation (AMETEK) at room temperature. The frequency range of EIS was 100 kHz–0.1 Hz. The CV curves were obtained at a rate of 0.1 mV s$^{-1}$ in a voltage range of 3.0–4.6 V (vs. Li/Li$^+$).

**Characterization of materials**. The SEM images of all prepared cathode materials and bare LCO particles were obtained using a field emission scanning electron microscope (FE-SEM, HITACHI SU8000) at 5 kV. For post-mortem analyses of SEI formation, the half-cells were disassembled carefully in the argon-filled glovebox to obtain positive electrodes of 2% LAF-LCO and bare LCO that was cycled for 5 and 100 cycles. After that, the cathodes were washed mildly with ethyl carbonate/diethylene carbonate (EC/DEC, volume:volume = 1:1) three times and dried fully under vacuum. To access the detailed morphologies of 2% LAF-LCO and bare LCO, a transmission electron microscope (TEM, HITACHI HT7700) was used at 120 kV. High resolution TEM (HRTEM) and selected-area electron diffraction (SAED) were conducted using a cold field emission transmission electron microscope (JEM-2100F) at 200 kV. Energy dispersive X-ray (EDX) mapping images of 2%LAF-LCO particles were also obtained using JEM-2100F.

Powder X-ray diffraction (XRD) patterns were obtained using X-pert Powder [PANalytical B.V., Cu Kα radiation (λ = 1.5406 Å), 40 kV, 40 mA] with a scan range of 10°–80°. The element analysis of modified LCO and bare LCO surfaces was performed by X-ray photoelectron spectroscopy (XPS, Escalab 250Xi). The data we obtained were corrected by a standard C 1s peak at 284.8 eV.

To confirm the presence of Co in the electrolyte solution after extensive cycling, cells of 2% LAF-LCO and bare LCO that were cycled for different cycles were harvested and disassembled in the argon-filled glovebox. All components of the cell were washed in 2 mL EC/DEC solvent for 5 d. Then, 0.5 mL of the solution was diluted in 2 mL HNO$_3$ (65–68%) and heated at 120 °C to evaporate the solvent. The white crystals that we obtained were collected and dissolved in 250 mL deionized water. The Co content was determined using inductively coupled plasma mass spectrometry (ICP-MS, XSENIES).

## Data availability

The data that support the findings of this study are available from the corresponding author upon request. The source data underlying Figs. 3a–d, 4a–d, 5a, b, 6a–c and Supplementary Figs 4, 5, 6, 7, 8, 9a, b, 10, 12, 13a–f, and 14a, b are provided as a Source Data file.

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

## Acknowledgements

This work was supported by the National Key R&D Program of China (grant 2018YFA0209600, 2016YFA0202900) and Natural Science Foundation of China (grant 21878268, 21676242). This research also used computational resources of the Texas Advanced Computing Center under contract no. TG-DMR170070. H.L.Z. thanks the start-up funds from Arizona State University.

## Author contributions

J.Q. and Y.L. conceived the idea, designed the experiments, and wrote the manuscript. J.Q. carried out the synthesis, electrochemical test, characterization, and analyzed the results with the help of J.Y. S.L. drew the 3D pattern to illustrate the synthetic process of the LAF-LCO. L.L., and H.L.Z. conducted the DFT calculations. X.W. provided help with editing this paper. We would like to thank Thomas A. Gavin, Professor Emeritus, Cornell University, for help in editing the manuscript.

## Additional information

**Competing interests:** The authors declare no competing interests.

