## [Peer Review File · Nature Communications]

Reviewers' comments:

Reviewer #1 (Remarks to the Author):

This paper reports a high voltage-type LiCoO₂ (LCO) cathode material, which was obtained by heat treatment of LCO particles in a mixed aqueous solution of Al(NO₃)₃•9H₂O, LiNO₃, and NH₄F, followed with a high-temperature annealing to form a modification layer consisting of Li, Al and F (LAF) on LCO surface. Such a LAF layer can prevent the direct contact of liquid electrolyte with LCO particles and suppress the phase transition of LCO at high voltages (>4.55V). As a result, the LAF-modified LCO cathode exhibits a high capacity (>173 mAh g⁻¹) and an improved cycleability compared to the untreated bare LCO cathode when cycled with a cut-off voltage higher than 4.55V. However, the present data can not support that the LAF-modified LCO is an applicable high voltage-type material for high energy density LIBs. The capacity retention of LAF-LCO cathode is only 83.5% in half-coin cells after 200 cycles and ~75% after 70 cycles when coupled with a graphite anode in full cells. This cycling performance is obviously insufficient for battery applications. In addition, the low Coulombic efficiency at cycling is another obstacle hindering its application in practical batteries. The average Coulombic efficiency of LAF-LCO cathode is only 99.64% after 5 cycles in half-coin cells and 99.17% in full cells at extended cycling. For these reasons, I cannot recommend this paper to be published in this journal.

Reviewer #2 (Remarks to the Author):

The manuscript here presented by Jiawei Qian et al. focused on the ternary Li, Al, F-modified LCO with stable and conductive interlayer by a facile and scalable hydrothermal assisted hybrid surface treatment. The authors confirmed that excellent electrochemical performance was achieved at high cut-off voltage of 4.6 V. This finding is quite impressive since it is generally acknowledged that the irreversible phase transitions occur at the voltage over than 4.5 V. However, in-depth analysis was lacked in this manuscript. It is very difficult to understand this finding, and what is new in mechanism. At the present stage, this manuscript might be not suitable for the publication prior to major modification.

The authors are suggested to address the following concerns.

1. The introduction section is repetitive and needs further refining.
2. line 47: "less" should be "more".
3. line 73: About capacity retention, please specify the number of cycles.
4. The format of the references is not standardized.
5. The mechanism is not carefully described, and additional experiment data is required so as to figure out the key factors.

Reviewer #3 (Remarks to the Author):

The paper is a well written account of a scalable method of producing stabilized electrodes for LIBs which. The material is well characterized and the conclusions regarding the composition and nature of the material are reasonable with significant support from DFT calculations. The paper and supplementary materials give a raft of information to support the authors' conclusions.

The choice of material system in this study is not in itself fundamentally new as AlF₃ has been used

previously to stabilize LCO. However, the method described here is new and appears readily tunable and to a large extent optimized. The performance of the devices, both coin and fuel cells, are impressive. The operation at >4.5 V seems to indicate that the disruptive phase transition is inhibited and Co loss prevented. Furthermore, the cycling stability is also sufficient.

In my opinion the subject matter and results reported in this manuscript would be of interest to the wider journal readership and not just specialists.

Note there is a typo in Line 159.

"We found that Reaction (6) is also exoergic whereas the formation energy (-0.06 eV) is much smaller," This should read Reaction (7)...

Reviewer #4 (Remarks to the Author):

A carefully executed study of a worthy subject has been reported extensively. Unfortunately many flaws of language, incomprehensible and/or garbled statement make complete understanding difficult. A careful and extensive revision is required before publication should be considered. For details see annotated manuscript.

Title: Electrochemical surface passivation of LiCoO₂ particles at ultrahigh voltage and their applications in advanced, safe lithium-based batteries

Dear Editors and Reviewers:

Thank you for your email on August 24th, 2018 regarding our manuscript. We greatly appreciate your effort, comments, and patience. In the revised version, we have made editorial changes according to your suggestions. Revised portions are highlighted in our new manuscript. The responses to the reviewers' comments are provided as follows:

Reviewers' comments:

Reviewer #1 (Remarks to the Author):

This paper reports a high voltage-type LiCoO₂ (LCO) cathode material, which was obtained by heat treatment of LCO particles in a mixed aqueous solution of Al(NO₃)₃•9H₂O, LiNO₃, and NH₄F, followed with a high-temperature annealing to form a modification layer consisting of Li, Al and F (LAF) on LCO surface. Such a LAF layer can prevent the direct contact of liquid electrolyte with LCO particles and suppress the phase transition of LCO at high voltages(>4.55V). As a result, the

LAF-modified LCO cathode exhibits a high capacity ($>173 \text{ mAh g}^{-1}$) and an improved cycleability compared to the untreated bare LCO cathode when cycled with a cut-off voltage higher than 4.55V. However, the present data can not support that the LAF-modified LCO is an applicable high voltage-type material for high energy density LIBs. The capacity retention of LAF-LCO cathode is only 83.5% in half-coin cells after 200 cycles and $\sim 75\%$ after 70 cycles when coupled with a graphite anode in full cells. This cycling performance is obviously insufficient for battery applications. In addition, the low Coulombic efficiency at cycling is another obstacle hindering its application in practical batteries. The average Coulombic efficiency of LAF-LCO cathode is only 99.64% after 5 cycles in half-coin cells and 99.17% in full cells at extended cycling. For these reasons, I cannot recommend this paper to be published in this journal.

Response: We thank the reviewer for the comment but we respectfully disagree with the referee as the cycling protocol of coin cell is different from that of commercial cells such as pouch cell. In commercial cells we usually age the cells via one charging process before cycling and during the aging process electrolyte will chemically react with electrodes and form a protective layer, which irreversibly consumes a certain amount of electrolyte. However, in a coin cell, we cannot do the aging process and the first few cycles, which render relatively low Coulombic efficiencies, are similar to the aging process in commercial cells. Therefore, a fair comparison to demonstrate the effectiveness of our modified electrode is to compare it with bare LCO. Firstly, the capacity retention of LAF-LCO improves 147% compared to bare LCO in half cells,

and average CE is improved from 98.86% to 99.64%. The full cells with LAF-LCO also demonstrate a high specific energy density of 600 Wh kg^{-1} at material level after 70 cycles while full cells with bare LCO only deliver 300 Wh kg^{-1} . Overall, compared to bare LCO, the cells with LAF-LCO show great improvement in electrochemical performance. Secondly, compared to the electrochemical performance of LCO using other strategies (previous reports by others), the cells with LAF-LCO also show the superiority. The LAF-LCO cells deliver an average CE of 99.64% (200 cycles), while LiAlO_2 coated LCO cells by ALD only show an average CE around 98% (50 cycles) according to the supplementary ref. 6. Third, the average CE not only depends on cathode materials, but also is associated with electrolytes and anode materials. To confirm that the improved electrochemical performance owes to the LAF modified LCO, we used the most common electrolyte (EC:DEC). In practical application, bare EC:DEC electrolyte is not suitable to high voltage cells charging up to 4.6 V because the EC:DEC electrolyte can decompose at high voltages and reduce the CE. However, our modified LCO cells can perform well even use EC:DEC. From the above reasons, our strategy shows its impressive superiority which should be considered on Nature Communications. In the future, we will take further research to realize the scale-up application of LAF-LCO.

Reviewer #2 (Remarks to the Author):

The manuscript here presented by Jiawei Qian et al. focused on the ternary Li, Al, F-modified LCO with stable and conductive interlayer by a facile and scalable hydrothermal assisted hybrid surface treatment. The authors confirmed that excellent electrochemical performance was achieved at high cut-off voltage of 4.6 V. This finding is quite impressive since it is generally acknowledged that the irreversible phase transitions occur at the voltage over than 4.5 V. However, in-depth analysis was lacked in this manuscript. It is very difficult to understand this finding, and what is new in mechanism. At the present stage, this manuscript might be not suitable for the publication prior to major modification.

The authors are suggested to address the following concerns.

1. The introduction section is repetitive and needs further refining.
2. line 47: “less” should be “more”.
3. line 73: About capacity retention, please specify the number of cycles.

Response: We thank the reviewer for the helpful comments. We have changed the statements according to the above three comments. We also include Prof. Thomas Gavin from Cornell University for English editing.

4. The format of the references is not standardized.

Response: We thank the reviewer for pointing this out. We have carefully checked the references and amended the nonstandard references.

5. The mechanism is not carefully described, and additional experiment data is required so as to figure out the key factors.

Response: Thank you for your constructive comment. We have done sufficient supplemental experiments to describe the mechanism of the effectiveness of such modified LCO. As mentioned in the manuscript, the irreversible phase transition over 4.55 V is mainly attributed to the larger internal strain, resulted from decreased Li ion diffusivity of the LCO, leading to larger Li ion concentration gradient. Meanwhile, as shown in Fig.11c in the Ref.11, the Li ion concentration gradient of surface structure is much larger than that of the bulk structure. Therefore, the key point is to increase Li ion diffusivity of surface structure in order to restrain the irreversible phase transition of LCO over 4.55 V. As shown in Supplementary Table 2 (structural parameters from the Rietveld refinements), the *c/a* ratio and unit cell volume of 2% LAF-LCO are larger than these of the bare LCO, while the difference is very small due to tiny amount of thin LACOF doping layer compared to the bulk LCO. The DFT calculation also shows that the *c*-lattice parameters of $\text{Li}_{1/3}\text{Al}_{1/3}\text{Co}_{2/3}\text{O}_{4/3}\text{F}_{2/3}$ at lithiated state and delithiated state are larger than these of LiCoO_2 . And the large *c*-lattice parameters of $\text{Li}_{1/3}\text{Al}_{1/3}\text{Co}_{2/3}\text{O}_{4/3}\text{F}_{2/3}$ can increase the Li diffusivity, undermine ion concentration gradients and lower the internal strain.

In addition, Figure S13 c and d show the *dQ/dV* curves of cells with bare LCO or 2% LAF-LCO electrodes after 1 cycle in the voltage range of 3.0-4.6 V (vs. Li^+/Li). The phase transition of LAF-LCO is similar to that of bare LCO, which indicates that the introduction of LACOF surface doping layer does not lead to additional phase transition. We also compare the phase transition of cells with modified LCO or bare

LCO before and after cycling via cycling voltammograms. Figure S11 e and f show the CV curves of the two cells before and after cycling for 50 cycles. In the case of bare LCO, the phase transition between the H1-3 host and O3 host disappears, while the same phase transition for LAF-LCO still remains after 50 cycles. It means that the introduction of LACOF doping layer does not change the intrinsic phase transition behavior of LCO, but restrains the harmful irreversible phase transition between the H1-3 and O3 host. The Figure S14 also supports the above conclusion.

Overall, the key factors are that Li, Al, F doping can increase c-lattice parameters and broaden Li^+ diffuse pathway during cycling, thus increase the Li diffusivity of the surface structure, undermine ion concentration gradients and lower the internal strain. Further, it retards the structure collapse caused by large internal strain. Finally, it makes the phase transition between the H1-3 and O3 host to be reversible.

Reviewer #3 (Remarks to the Author):

The paper is a well written account of a scalable method of producing stabilized electrodes for LIBs which. The material is well characterized and the conclusions regarding the composition and nature of the material are reasonable with significant support from DFT calculations. The paper and supplementary materials give a raft of information to support the authors' conclusions. The choice of material system in this study is not in itself fundamentally new as AlF_3 has been used previously to stabilize

LCO. However, the method described here is new and appears readily tunable and to a large extent optimized. The performance of the devices, both coin and fuel cells, are impressive. The operation at >4.5 V seems to indicate that the disruptive phase transition is inhibited and Co loss prevented. Furthermore, the cycling stability is also sufficient.

In my opinion the subject matter and results reported in this manuscript would be of interest to the wider journal readership and not just specialists.

Response: We thank the referee for the thoughtful review of our manuscript and are happy to hear that this is an attractive work with high novelty and significance for publication on Nature Communications. We have carefully revised our manuscript based on your recommendations. We also detailed the responses for your specific comments below.

Note there is a typo in Line 159. "We found that Reaction (6) is also exoergic whereas the formation energy (-0.06 eV) is much smaller," This should read Reaction (7)...

Response: We thank the reviewer for pointing out this error. We have carefully checked it and corrected the mistake.

Reviewer #4 (Remarks to the Author):

A carefully executed study of a worthy subject has been reported extensively. Unfortunately many flaws of language, incomprehensible and/or garbled statement make complete understanding difficult. A careful and extensive revision is required before publication should be considered. For details see annotated manuscript.

Response: Thank you for your careful reading of our manuscript. We have carefully checked and corrected the mistakes according to the annotated manuscript. We also include Prof. Thomas Gavin from Cornell University for English editing.

We have carefully proofread the manuscript and corrected some other mistakes according to your suggestions.

A revised manuscript is enclosed. Please feel free to contact us if you have any further questions.

Best regards,

Yingying Lu, Ph.D.

College of Chemical and Biological Engineering

Zhejiang University

Hangzhou, Zhejiang, 310027, P. R. China

Reviewers' comments:

Reviewer #1 (Remarks to the Author):

In the response, the authors ascribe the poor cycling performance and low coulombic efficiency of LAF-modified LCO cathodes to lack of an age process, which is usually used to stabilize the electrochemical performance of commercial lithium ion cells. If this is a reason, the authors can also do it by charging one cycle prior to cycling the coin cells. But the authors now tell us that the age process can not be done for a coin cell. I do not agree with the authors' response and still think that the present data can not sufficiently support the conclusions.

Reviewer #2 (Remarks to the Author):

The manuscript has been improved according to the comments. It should be noted that there is still big space to investigate the mechanism theoretically and experimentally. Since the experimental finding is very attractive, my personal opinion is to recommend for the publication.

Reviewers' comments:

Reviewer #1 (Remarks to the Author):

In the response, the authors ascribe the poor cycling performance and low coulombic efficiency of LAF-modified LCO cathodes to lack of an aging process, which is usually used to stabilize the electrochemical performance of commercial lithium ion cells. If this is a reason, the authors can also do it by charging one cycle prior to cycling the coin cells. But the authors now tell us that the aging process can not be done for a coin cell. I do not agree with the authors' response and still think that the present data can not sufficiently support the conclusions.

We thank the reviewer for the comment which pointed out the inappropriate statements. We consent that the aging process is indeed available for coin cells. However, a very important factor that should not be ignored, is the degassing process which is usually used after aging process for commercial lithium ion cells. In the process for commercial pouch cells, the pocket is usually reserved for large amount of organic gas (CO_2 , C_2H_4 , etc) which is generated during the aging process. After the aging process, the pocket will be cut off and the organic gas which is harmful to electrochemical performance of pouch cells will be released. However, in the coin cells, due to the specific structure of the coin cells, there is lack of effective methods to release the large amount of harmful organic gas generated from the initial charging process. In the subsequent cycling process, the remained organic gas will continue to make negative influence on electrochemical performance of coin cells. Above all, we agree with the points by the reviewer that the aging process is available to coin cells, however, due to lack of effective degassing process, the coin cells showed inferior electrochemical performance. On the other hand, the coin cells usually cycle 1-2 cycles at low current density as simplified aging process by most researchers. Therefore, the initial coulombic efficiency is lower than average coulombic efficiency. The initial coulombic efficiency (90.6%) of bare LCO is similar with other research results, such as 90.7% of bare LCO (Supplementary 6) and 90.83% of bare LCO (Supplementary 7).

In sum, we thank the reviewer for the constructive comment. Meanwhile, the experimental results also showed the superiority of our study. Of course, we noted that there is still big space to improve the electrochemical performance experimentally. We will take steps in synthesis process, coating process, aging and degassing process for pouch cells in the future study, to further improve the electrochemical performance.

REVIEWERS' COMMENTS:

Reviewer #1 (Remarks to the Author):

I satisfy with the author's response and now agree with the paper to be accepted for publication.

Reviewer #1 (Remarks to the Author):

I satisfy with the author's response and now agree with the paper to be accepted for publication.

Response: Thank you very much for your positive comment to our manuscript. We really appreciate your careful review.